# Evaluation of Factors Contributing to the Effectiveness of Internal Audit Quality in Pakistani Commercial Banks

## Madiha Afzal

School of Accounting, Shanxi University of Finance and Economics, Taiyuan 030006, China;
afzalsandhu431@yahoo.com

**Abstract:** The Pakistani banking sectors facing numerous challenges because of poor internal audit quality. Internal audit quality has long been a source of contention. The current study examines the factors that affect internal audit quality in Pakistani commercial banks. Internal audit quality evaluated through potential factors such as competence, objectivity, performance, board audit committee support, and independence. Along with these factors, a questionnaire designed to determine the nature of the problems confronting Pakistani commercial banks. 102 questionnaires disseminated among the chief internal auditor, chief financial officer, board audit committee, and managers of 26 listed commercial banks. The impact of the factors on internal audit quality investigated using a binary logit regression model and multiple correspondence analyses. Findings show that performance, competence, and objectivity factors are statistically positively significant that influenced internal audit quality to improve it. This research helps improve the internal audit quality in Pakistani commercial banks.

**Keywords:** audit quality effectiveness; Pakistan commercial banks; logit modeling

## 1. Introduction

The internal audit requires more strength due to rapid advancement in the global economy, technology, and complexity of rules and regulations in the last decades that set towards new directions, which ultimately support the management and added value for the organization (Wang et al. 2022; Ado et al. 2020). The strength requires more skillful internal auditors who performed their best in leading the global economy (Alsabti and Khalid 2022; Vashdi et al. 2022). Along with internal auditor skills and education, in the current scenario, more attention needs to make sure effectiveness of internal audit quality (George et al. 2015). Undoubtedly, the effectiveness of internal audit quality is a big challenge that can be promptly overcome with the help of key elements of good governance (Ahmet 2021). So that more important in the current context of the study to describe the internal audit quality effectiveness and determine the factors that can add value to internal audit (Abdelrahim and Al-Malkawi 2022; Prasad et al. 2022).

Internal audit effectiveness is mandatory to achieve the set goals. The effectiveness of internal audit quality is a set goal for evaluation of either audit function is capable to achieve the set goal by the organization (Turetken et al. 2020). An organization that set effective internal audit function are more reliable and performs well, estimate future risk, and can take appropriate action against any risk and it helps to better improvement will last (Alqudah et al. 2019). Furthermore, the internal audit quality function helps to achieve public confidence that works as an asset for any organization which is achievable through reliable financial reporting and these are basic elements that every organization must contain (Phan et al. 2020). Owing to these elements, an organization can achieve competent internal audit staff and a leadership board audit committee whose ultimate objective is to achieve effective internal audit quality. Due to rapid advancement in audit systems, the effectiveness of internal audit quality becomes a more fruitful topic in the last

decades. Because organizational success and survival depend on the internal auditor's best performance (Badara and Saidin 2013). Internal auditors have one of the basic duties to check daily routines and solve problems. This day-to-day check helps the internal auditor to cater to risk assessment at an advanced level (El Gharbaoui and Chraibi 2021). The big challenge of internal auditors is the lack of authority, responsibility, and scope, which is not mentioned by the higher authority owing to this internal audit departments could not understand the real internal audit function (Al Nuaimi et al. 2020). Furthermore, still exist challenges and expectations from internal auditors higher to operate daily activities at their level best. So, the basic attributes of internal audit are not designed (Anojan 2022).

Therefore, auditor performance affecting. Because the achievement of set goals is only possible if the performance and other attributes are well-designed and mentioned (Potjanajaruwit 2022). The internal audit function is only considered effective if it can achieve the target on time (Al-Okaily et al. 2022). Owing to these issues and challenges that persist in the audit system attract more attention to the researcher. So, it's mandatory to find that variable that is necessary to make internal audit quality effective (Hoai et al. 2022).

Financial institutions have been under a strong and critical public spotlight in the last decades due to the rapid trend in internal control failure and risk management faced by both developing and developed world (Otache et al. 2022; Muñoz-Izquierdo et al. 2019). Besides many issues and challenges still exist in the Pakistani banking sector another fact is that there is no denying that in the last decade, there is a lot of progress made such as progress in risk management, improvement in governance practices, launching new products and services, updated technologies, and better customer services. Internal audit effectiveness has greatly associated with the internal audit because the ultimate objective is to achieve desired objectives of the organization which is to attain error-free audit activities (Khatib et al. 2021; Li 2022; Le et al. 2022).

For more understanding have a look auditing system in Pakistani banks will provide clear insight

Internal auditing system in Pakistani commercial banks

Pakistan as a developing country is much interested to explore new methods to implement that help to achieve internal audit effectiveness since 1973. In this regard, the development of internal audits in Pakistani commercial banks is maintained by the Pakistan Institute of Internal Auditors (IIA). The main purpose of this institute is to develop a specific charter namely Internal Audit Charter (IAC). In this charter mentioned the role, duties, responsibilities, and who reports to whom. The IAC is checked and reviewed by the Board Audit Committee (BAC), this IAC is updated with time and situation under the guidance of BAC.

IAC, IAC must cover the following dimensions to achieve internal audit effectiveness:

(a)  Internal audit function must involve formal standing, responsibilities, powers and authorities with the help of international best practices, guidelines, and standards etc.
(b)  To maintain the internal audit effectiveness, the internal control environment must be effective for the internal auditor openly and independently express their opinion on different affairs.
(c)  Chief internal auditors, auditors, and managers must perform day-to-day roles and responsibilities and proper check and balance on every activity while performing audit.
(d)  Internal auditors have free access to banks records, data, files, information, meeting, properties, and people.
(e)  According to the nature of the problem and situation IAF provided consultancy if required.
(f)  Independence of internal auditors depends on the IAF that is set by the organization.
(g)  The basic report system of who reports to whom in IAF provides clear insights to BAC and related members, such as internal stakeholders.
(h)  In case of external suggestions, opinion if require it's all clearly defined to what extent is needed mentioned in IAF.

(i)  In this charter also defined to what extent CIA evaluation mechanisms are included.

(j)  It is mandatory that assurance of IAF must be assessed by IIA standards and periodically updated with time and situation.

Due to abortive internal audit quality, a lot of studies contribute to internal audit quality effectiveness. But few studies contribute from Pakistan. There is no denying that the banking sector contributes the best to Pakistan's economy, but due to abortive audits faces many challenges, such as irregularities and fraud from the top level to the lower level. The Internal audit helps to overcome these challenges and improve the overall governance structure, which is the main objective of the current studies to eliminate fraudulent and corrupt activities from banking sectors. The current research will contribute to improving the audit department in the banking sector and prevent all those fraudulent activities with the help of factors and give insight to Chief internal auditors to keep in mind these factors while conducting internal audit activities. In this way, the overall governance structure of the audit department will improve in Pakistani commercial banks. Furthermore, the scholars could not find any specific literature in Pakistan which used current factors and their contribution towards internal audit quality effectiveness. So current studies' contribution will highly appreciate in this regard.

The remaining paper is classified as follows Section 2 involves the background of the study Section 3 is research methodology Section 4 is the results of the study based on model development and Section 5 conclusion.

## 2. Background of the Study and Research Hypotheses

### 2.1. Effectiveness of Internal Audit Quality

Due to the complexity of internal audit effectiveness, it has been less studied in accounting and finance. Internal audits are defined by different authors in different ways. Internal audit effectiveness helps to achieve the set target by the organization and to reach the goals within a specific time is the effectiveness of internal audit quality (Hazaea et al. 2020a, 2020b, 2020c). According to the basic definition, it is an independent assurance of objectives and provides consulting that is designed to add value and improvement of the organization's current operations (Lenz et al. 2018). So, IAF considers effective for the organization when it helps an organization to add value and creativity in current activities (Ta and Doan 2022).

Effective internal control, governance system, and risks management assessment are basic tools that help to evaluate the effectiveness of internal audit quality function by the chief financial officer, management, and board audit committee claimed by the previous literature, (Pooe et al. 2022; Jung and Cho 2022) their studies provide clear insights that internal audit could only add value and creativity to the organization when achieving ultimate objective of economic, effective internal control that is implied by the chief internal auditor with the support of management.

Internal audit effectiveness is a debatable issue over the decades. The effectiveness of internal audit quality is set goals by the organization (Joshi 2022). According to the definition of internal audit, the main objective is to add value to the organization. So that internal audit effectiveness can be easily measured if it actual add value for the organization.

For the current study, factors affecting internal audit effectiveness are because Pakistani commercial banks could not have an effective internal control system, regulatory compliance, and customer services. So that banks cannot achieve their economic and social objectives. Economic objectives are to achieve desired target finance. For this purpose, an effective internal audit quality system is mandatory because all the financial reports evaluate by the auditor. After a deep overview of previous studies, the following five factors affecting the effectiveness of internal audit quality.

### 2.2. Research Hypothesis

After insight and reviewing the previous literature specifically in audit accounting and finance, the scholars have performed their best efforts to fill it by developing the research

hypothesis according to the nature of the problem in the current situation of their countries. The scholars developed and formulated the hypothesis to achieve the best and broad set of objectives. According to Pakistani commercial banks' perspective, there are five factors affecting internal audit effectiveness. The research seeks to test the following hypothesis:

### 2.3. Competence of Internal Auditor

The competence of an internal auditor involves knowledge, experience, and skills. These attributes are mandatory for an internal auditor to fulfill his duties. These duties ultimately help to achieve effective internal audit quality. Competence of the internal auditor factor works as the main pillar in the audit system. Without these attributes, the internal auditor cannot perform their professional activities. Professionalism first entity must know according to require fields. Furthermore, to perform their internal audit activities auditors must have good communication skills, so that communicate with other members of the department to achieve the desired objective in the banking system. Auditors must have critical thinking to predict future risks and make reforms in such a way as to overcome risks in the future. Competency of the auditor to help work and collaborate with different internal departments to achieve effective internal audit quality. According to (Singh et al. 2021) and the educational level of the internal auditor, the internal auditor operates the procedures with professional experience and knowledge are a basic pillar to evaluate the competence of internal auditor. According to previous studies performed by different authors in different countries current study is designed following hypothesis:

**H1.** *There is a positive relationship between competences of internal auditor the effectiveness of internal audit quality.*

### 2.4. Performance of Internal Auditor

As with other factors, the performance of the internal auditor is essential. Auditor work has many challenges due to the complexity of the job. The performance of the internal auditor requires covering all issues in current scenarios that are facing the banking system. The basic purpose of an auditor is to present fair and clear audit reports, to attract internal and external performance. It is only possible when auditor performs their duties on time. The performance of the internal auditor works as a pillar for any organization because if there is one mistake the whole process of the organization will be affected (Lenz and Hahn 2015). So, the performance of the organization can be considered complete if it helps to achieve the organization's set goals and objectives. Organization effectiveness depends ultimately on auditor performance. It is only possible if auditor goals match with organization goals and objectives. This is due that auditor perform their duty well. So that according to the importance of performance factors, the second hypothesis of the current study is as follows:

**H2.** *There is a positive relationship between the performances of internal auditors and the effectiveness of internal audit quality.*

### 2.5. Independence of Internal Auditor

The independence of internal auditors means performing their duties unbiasedly. To achieve desired objectives, it is mandatory to give clear instructions to the Chief internal auditor to branch managers there is no involvement while performing audit activities. These restricted rules must be set by the auditors as well. Because based on internal auditor reports internal and external investors to attract towards organization. If it is made unbiased it is very convenient for the investors in the future for the reliability of the reports. It is only possible if there is a strong connection among internal audit departments. Internal audit departments must have strong communication (Alqudah et al. 2019). The audit department owing to its professionalism detects the errors so quickly. And if the auditor performs their duties unbiasedly there is no threat to external auditor. Because if an

organization needs external auditors, then we must need to check internal auditor progress during the annual meeting Dellai and Omri (2016). This is due to the auditor having all the rights to check and evaluate each instrument without any restrictions and involvement of other authorities. According to previous studies performed by different authors in different countries current study is designed following hypothesis:

**H3.** *There is a positive relationship between the independence of internal auditors and the effectiveness of internal audit quality.*

### 2.6. Objectivity of Internal Auditor

Objectivity is a key factor for the internal auditor. If the auditor works, their activities objectively and according to the structure then the organization progress ultimately. As aforementioned importance clearly defined the factor importance in the current study. In the last decades audit system rapidly requires more attention due to their importance in the current progressive environment. According to (Kassie 2021) previous studies and researchers found that the objectivity factor is closer to the internal audit function. According to others argue their work it is as important as blood in the body. Fair and clear reports designed by the auditor if the auditor's objectives and organization's objectives same. And then the organization can achieve the ultimate desired objectives.

**H4.** *There is a positive relationship between objectivity of internal auditor the effectiveness of internal audit quality.*

### 2.7. Board Audit Committee Support

The organization's daily routine depends on the manager's activities. And managers are always ready to adopt new methods and techniques to update daily activities. Managers and internal auditor support are mandatory for organizational goals (Nguyen 2022). Internal audit activities cannot perform accurately without management's strong support. Audit activities always perform based on providing information by management (Nguyen 2020). According to strong management support, a board audit committee, and a chief financial officer are the basic pillar that can help to achieve organizational desired goals. Internal audit departments need to build strong communication with the board audit committee, and management. Because if they have good communication then they can achieve an effective internal control system. Effective internal control and effective internal audit are mandatory to achieve the financial performance of the organization Umar and Dikko (2018).

**H5.** *There is a positive relationship between board audit committee supports of internal auditing and the effectiveness of internal audit quality.*

According to the nature of issues confronted by Pakistani commercial banks hypothesis developed, and the basis on the abovementioned hypothesis conceptual framework constructed in Figure 1.

This section may be divided into subheadings. It should provide a concise and precise description of the experimental results, their interpretation, as well as the experimental conclusions that can be drawn.

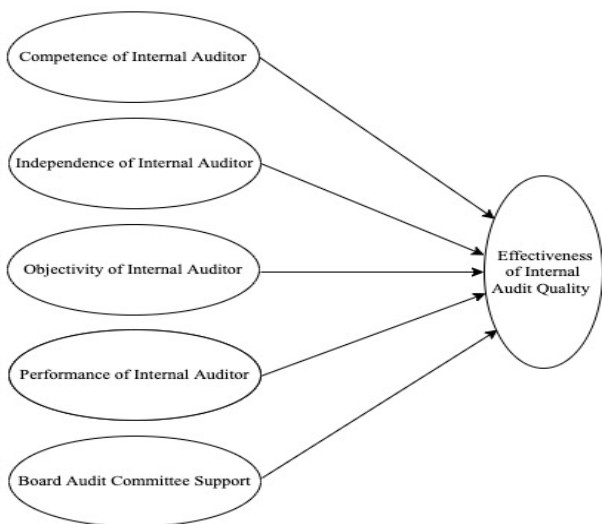

**Figure 1.** Conceptual framework.

## 3. Research Methodology

### 3.1. Research Data

The objective of the current study is to evaluate factors affecting the internal audit quality effectiveness of Pakistani commercial banks. To evaluate the alliance between dependent and independent variables explanatory research was used (Filfilan 2022). Explanatory research is an approach to examine the data received from respondents via a ques-tionnaire. The basic purpose to use explanatory kind of research is to evaluate, explain and forecast the alliance between variables via a statistical test namely regression analysis (Gros et al. 2017).

### 3.2. Research Approach

The deductive research approach is used because the conceptual framework develops based on previous studies and testing hypotheses. This approach helped to identify the relationship among the factors (Enekwe et al. 2020). For testing the hypothesis quantitative method was used. Source of data used primary source data. The advantage of primary source data is to provide firsthand data without interpretation, opinion, and analysis of others.

### 3.3. Sample and Data Collection

In the current study, primary data was collected using a survey method to determine the effectiveness of internal audit quality in Pakistan's commercial banks. The population consisted of 26 listed commercial banks in Pakistan, and geographic segmentation was used to select the target sample. Before sending the questionnaire, to communicate first send a consent letter to the Chief internal auditor who attains access to email from other auditors. A total of 102 questionnaires were disseminated among internal auditors of commercial banks, and 92 valid responses were obtained, resulting in a response rate of 93%. The questionnaire was designed and structured to be relevant to the current study and included questions related to internal auditor competency, objectivity, independence, performance, and board audit committee support. The Likert scale was used to evaluate the responses to each question. The survey method was deemed appropriate for this study as it allowed for accurate and comprehensive data collection from a representative sample of internal auditors in Pakistani commercial banks.

### 3.4. Research Model Development

Logit models are commonly used for binary outcome variables that can only take two possible values (e.g., agreement/disagreement, yes/no), but extensions of the logistic

regression model can also be used for ordinal outcome variables measured on a Likert scale. The ordinal logistic regression model estimates the cumulative probabilities of the outcome being in each category (e.g., strongly agree, agree, neutral, disagree, strongly disagree) or a lower one, assuming the odds of being in a higher category versus a lower one is proportional across all categories. The coefficients represent the log odds of the outcome being in a higher category versus a lower one given a one-unit increase in the predictor variable while holding all other variables constant. Exponentiating the coefficients yields odds ratios, which represent the multiplicative effect of a one-unit increase in the predictor variable on the odds of being in a higher category versus a lower one.

The data collected for the current study was on a Likert scale and analyzed using descriptive statistics. The analysis revealed a higher number of responses in two categories: "agree" and "strongly agree." To observe the patterns in the data, we combined the "agree" and "strongly agree" responses into one category, and the "neutral" to "strongly disagree" responses into another category. As a result, we employed a binary logit regression model for our research.

### 3.5. The Logit Model

The logistic formulas are stated in terms of the probability that Y = 1, which is referred to as $p$. The probability that $Y$ is 0 is 1–$p$. To test the research hypothesis, the following model is formulated:

$$logit(P(Y = 1)) = log[P(Y = 1)/P(Y = 0)]$$
$$= \beta 0 + \beta 1 X1 + \beta 2 X2 + \beta 3 X3 + \beta 4 X4 + \beta 5 X5$$

where $p$ is the probability of the binary outcome, $\beta 0$ is the intercept term, $\beta 1$ to $\beta 5$ are the coefficients for predictors $X1$ to $X5$, and logit () is the logit function which is the natural logarithm of the odds of the outcome.

Where,

$\beta 0$ = constant.

$logit(p)$ = estimates the log-odds of agreement towards effectiveness of internal audit quality.

$X1$ = Competence of internal auditor.

$X2$ = Independence of internal auditors.

$X3$ = Performance of internal auditor.

$X4$ = Objectivity of internal auditor.

$X5$ = Board audit committee support.

## 4. Results of the Study

Total 26 listed commercial banks in Pakistan. The hierarchy of internal auditors in these banks is 150 internal auditors' teams available in each bank. This team periodically sends it to different branches of banks. The team is consisting basically of 5 members. These members ranked as senior auditor, junior auditor, and trainee as well.

Table 1 provides the demographic information of the respondents, including the bank name, frequency, and percentage of respondents from each bank. It appears that the study includes a total of 92 respondents from 27 different banks in Pakistan. The number of respondents from each bank ranges from 1 to 5, with most banks having 4 respondents each. This table can be useful for understanding the distribution of respondents and ensuring that the sample is representative of the banking industry in Pakistan.

Table 2 shows the qualification demographic information of the respondents. Out of 91 participants, 30 (33.0%) held a Bachelor's degree, 24 (26.4%) held a Master's degree, 17 (18.7%) held an ICMA (Institute of Cost and Management Accountants) qualification, 8 (8.79%) held an ACCA (Association of Chartered Certified Accountants) qualification, 4 (4.40%) held a CPA (Certified Public Accountant) qualification, 4 (4.40%) held a PIPFA (Pakistan Institute of Public Finance Accountants) qualification, 2 (2.20%) held a CA (Char-

tered Accountant) qualification, and 2 (2.20%) held a CIA (Certified Internal Auditor) qualification.

**Table 1.** Respondents' demographic information.

| Bank Name | Frequency | Percentage |
|---|---|---|
| Al Baraka | 1 | 1.09 |
| Allied Bank Limited | 4 | 4.35 |
| Askari Bank | 4 | 4.35 |
| Bank Al Habib | 4 | 4.35 |
| Bank Alfalah | 4 | 4.35 |
| Bank Islamic | 1 | 1.09 |
| Barclays Bank | 3 | 3.26 |
| Citibank Pakistan | 3 | 3.26 |
| Faysal Bank | 4 | 4.35 |
| First Women Bank | 4 | 4.35 |
| Habib Bank Limited | 4 | 4.35 |
| Habib Metropolitan Bank | 1 | 1.09 |
| HSBC Pakistan | 4 | 4.35 |
| JS Bank | 5 | 5.43 |
| KASB Bank Limited | 4 | 4.35 |
| MCB Bank | 4 | 4.35 |
| Meezan Bank | 4 | 4.35 |
| Maybank | 3 | 3.26 |
| National Bank of Pakistan | 4 | 4.35 |
| NIB Bank Pakistan | 4 | 4.35 |
| Royal Bank of Scotland | 3 | 3.26 |
| Silkbank Limited | 3 | 3.26 |
| Soneri Bank | 4 | 4.35 |
| Standard Charted Bank | 4 | 4.35 |
| Summit Bank | 4 | 4.35 |
| The Bank of Punjab | 1 | 1.09 |
| United Bank Limited | 4 | 4.35 |

**Table 2.** Qualification demographic information.

| Qualification | Frequency | Percentage |
|---|---|---|
| Bachelor | 30 | 33.0 |
| Master | 24 | 26.4 |
| CA | 2 | 2.20 |
| CIA | 2 | 2.20 |
| CPA | 4 | 4.40 |
| ICMA | 17 | 18.7 |
| PIPFA | 4 | 4.40 |
| ACCA | 8 | 8.79 |

Table 3 shows the level of experience demographic information, with 25% of respondents having above five and less than ten years of experience, 34.8% having above two less than five years of experience, 35.9% having less than two years of experience, and only 4.35% having above ten years of experience.

**Table 3.** Level of experience demographic information.

| Level of Experience | Frequency | Percentage |
|---|---|---|
| Above five less than ten years | 23 | 25.0 |
| Above ten years | 4 | 4.35 |
| Above two less than five years | 32 | 34.8 |
| Less than two | 33 | 35.9 |

Table 4 shows the specialization demographic information, with 33% of respondents specializing in accounting and finance, 26.4% specializing in business administration, and 8.79% specializing in management. For questions responses from respondents have a look at Figures 1–6.

Figure 2 responses from respondents indicate that 92% strongly agree to agree that auditor educational level, training must be ensured other 3% only strongly disagree to disagree. According to the questionnaire settlement, it was the first section included six questions and every question response showed a positive and significant impact on the audit quality effectiveness.

From Figure 3, the objectivity factor same as the first factor has six questions and it was added in the second section of the questionnaire of the responses, 91% strongly agree to agree the other ratio 4% disagree to strongly disagree the actual response showing in this figure of each question and indicates this factor also has positive influence towards audit effectiveness internal audit quality.

**Table 4.** Specialization demographic information.

| Specialization | Frequency | Percentage |
| --- | --- | --- |
| Accounting and finance | 30 | 33.0 |
| Business administration | 24 | 26.4 |
| Management | 8 | 8.79 |

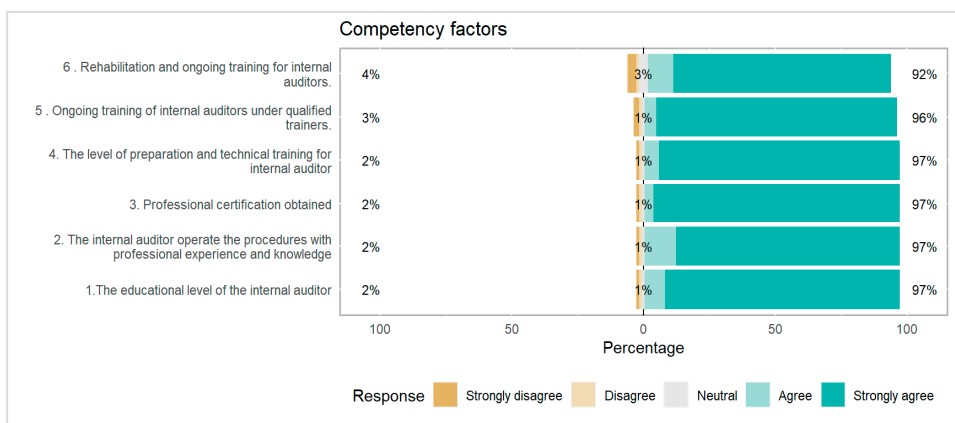

**Figure 2.** Competency factors determining the internal audit performance.

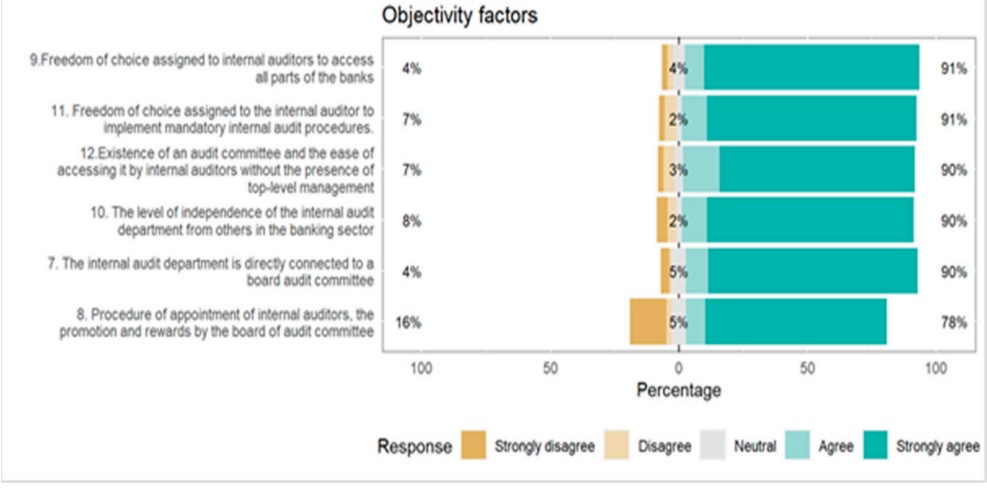

**Figure 3.** Objectivity factors determining the internal audit performance.

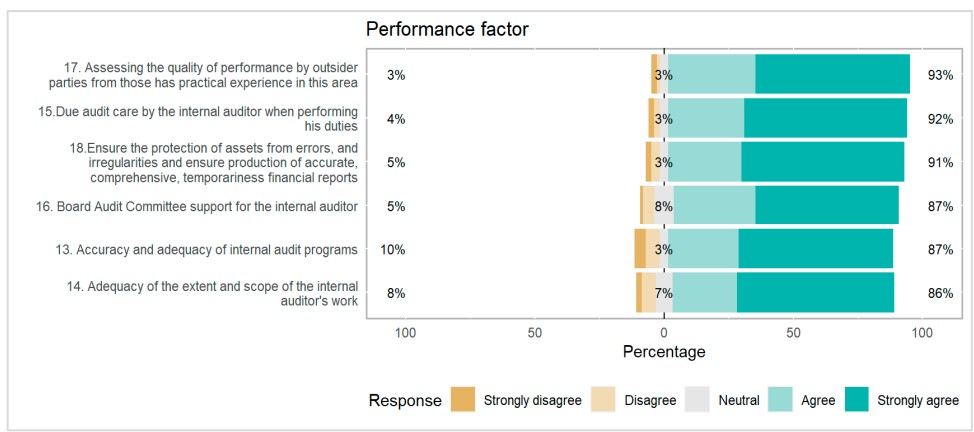

**Figure 4.** Performance factors determining the internal audit performance.

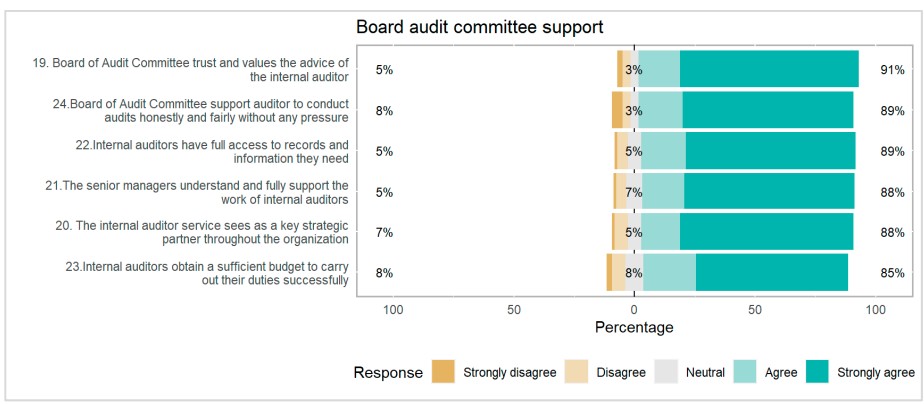

**Figure 5.** Board Audit Committee Support factors determining the internal audit performance.

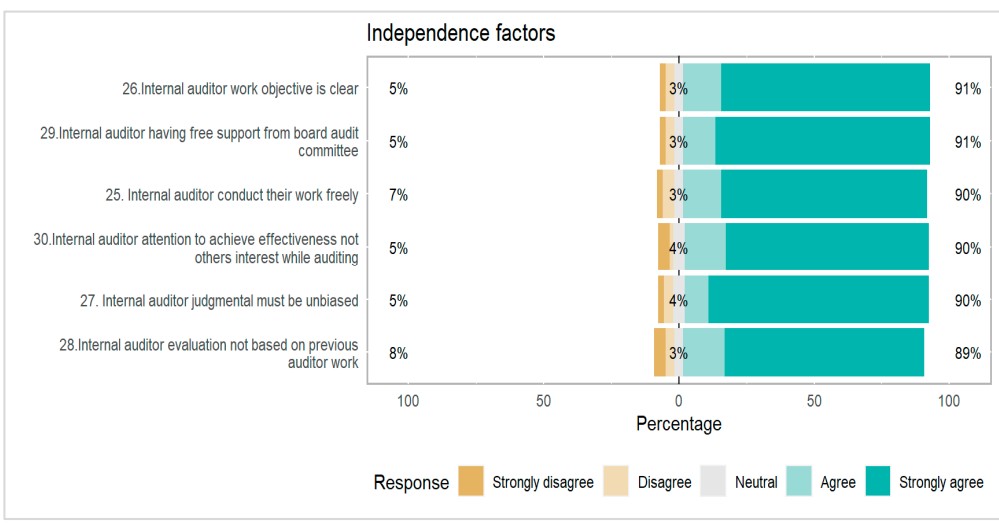

**Figure 6.** Independence factors determining the internal audit performance.

The performance factors responses presented in Figure 4, has indicating responses such as 93% is the highest ratio of strongly agree to agree and every question in all figures has a positive and significant influence on internal audit quality. So that strongly disagree, disagree, and neutral responses are 7% to 3% as well.

The board audit committee support factor presented in Figure 5 involves a 91% to 85% ratio as a response rate it was a good response rate because evaluating things are easier due

to the highest response towards strongly agree to agree and other such as strongly disagree to disagree, and neutral ratios are 3% to 8% are not worth as per sample population. Board audit committee support also has a positive influence on internal audit effectiveness.

In Figure 6, the independence factor involves a 91% to 89% ratio as a response rate it was a good response rate because evaluating things are easier due to the highest response towards strongly agree to agree and other such as strongly disagree to disagree, and neutral ratios are 3% to 4% are not worth as per sample population. The independence factor is the last section of the questionnaire as an independent variable with a positive and significant influence on internal audit effectiveness.

Internal audit quality last section as the dependent variable all other variables are evaluated based on this factor in Figure 7. The question was designed as per auditor instructions and guidelines. The ratio shows that 95% to 89% strongly agree to agree and others strongly disagree to disagree and neutral are 1%, 2%, and 4% which means all the factors need more attention to check and take appropriate measure by the Chief internal auditor for better performance by the internal auditor and other will ultimately be updated.

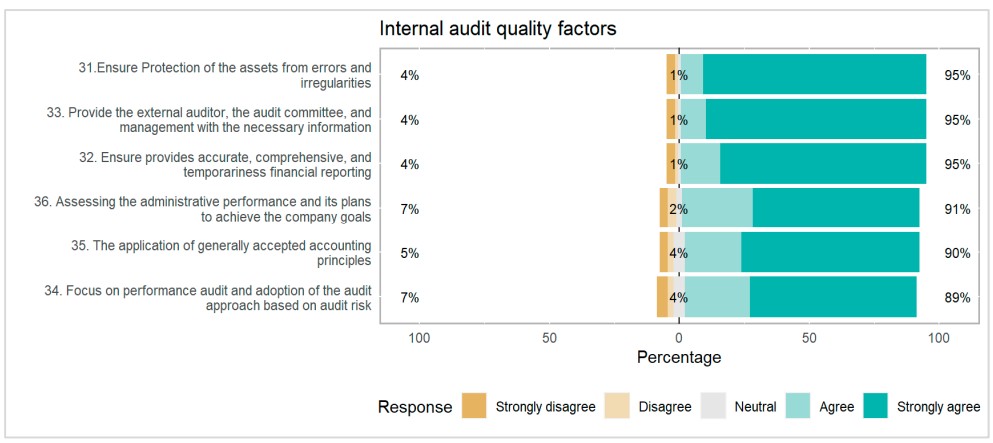

**Figure 7.** Internal audit quality factors determining the internal audit performance.

Presented Table 5 shows Cronbach's alpha coefficient with a 95% confidence interval (CI) for each factor, as well as for the overall scale. Cronbach's alpha is a measure of internal consistency or reliability, which assesses how well items in a scale or questionnaire are measuring the same construct.

**Table 5.** Reliability analysis of different factors.

| Factors | Cronbach's $\alpha$ with 95% CI |
| --- | --- |
| Competency | 0.79 (0.70–0.85) |
| Objectivity | 0.79 (0.56–0.89) |
| Performance | 0.88 (0.83–0.92) |
| Board audit committee support | 0.85 (0.77–0.90) |
| Independence | 0.91 (0.80–0.95) |
| Internal audit quality | 0.88 (0.73–0.95) |
| Overall | 0.93 (0.89–0.95) |

The results indicate that all factors have acceptable levels of internal consistency, as all Cronbach's alpha coefficients are above the commonly accepted threshold of 0.70. Specifically, Competency and Objectivity have a Cronbach's alpha of 0.79 with 95% CIs of 0.70–0.85 and 0.56–0.89, respectively. Performance has the highest Cronbach's alpha of 0.88 with a 95% CI of 0.83–0.92. Board audit committee support and Independence have

Cronbach's alpha coefficients of 0.85 and 0.91, respectively, with 95% CIs of 0.77–0.90 and 0.80–0.95. Finally, internal audit quality has a Cronbach's alpha of 0.88 with a 95% CI of 0.73–0.95. The overall scale has a high level of internal consistency, with a Cronbach's alpha coefficient of 0.93 and a 95% CI of 0.89–0.95. This suggests that the items in the scale are measuring the same construct and are reliable in assessing the quality of internal audits see for, (Cohen and Sayag 2010).

Based on the presented Table 6, the univariate analysis shows that the odds of the response variable being in the agreement category are significantly higher for performance, competence, and objectivity, compared to the disagreement category. For example, the odds of being in the agreement category for performance are 15.867 times higher than for disagreement (OR = 15.867, 95% CI = 4.641–54.242, $p < 0.001$).

**Table 6.** Factor associated with agreement of internal audit quality using univariate and multivariable logit-models.

| Factor | Response | Univariate Analysis | | Multivariable Analysis | |
|---|---|---|---|---|---|
| | | Crude OR (95% CI) | *p*-Value | adj. OR (95% CI) | *p*-Value |
| Performance | Disagreement | 1.00 (Ref.) | | 1.00 (Ref.) | |
| | Agreement | 15.867 (4.641, 54.242) | <0.001 | 15.836 (2.827, 88.704) | 0.0017 |
| Competence | Disagreement | 1.00 (Ref.) | | 1.00 (Ref.) | |
| | Agreement | 6.667 (2.565, 17.327) | <0.001 | 7.351 (2.109, 25.625) | 0.0017 |
| Objectivity | Disagreement | 1.00 (Ref.) | | 1.00 (Ref.) | |
| | Agreement | 7.526 (2.369, 23.907) | <0.001 | 7.213 (1.389, 37.452) | 0.0187 |
| Board audit committee support | Disagreement | 1.00 (Ref.) | | 1.00 (Ref.) | |
| | Agreement | 1.743 (0.634, 4.789) | 0.2812 | 2.166 (0.416, 11.267) | 0.3583 |
| Independence | Disagreement | 1.00 (Ref.) | | 1.00 (Ref.) | |
| | Agreement | 0.516 (0.153, 1.74) | 0.2863 | 0.11 (0.013, 0.942) | 0.0439 |

The multivariable analysis, which adjusts for potential confounding variables, shows similar results. After adjusting for potential confounders, the odds of being in the agreement category are still significantly higher for performance, competence, and objectivity. For example, the adjusted odds ratio (adj. OR) for performance is 15.836 (95% CI = 2.827–88.704, $p = 0.0017$), indicating that even after controlling for potential confounding variables, the odds of the response variable being in the agreement category are about 15.8 times higher for performance than for disagreement.

However, the factor board audit committee support does not show a significant association with the response variable in either the univariate or multivariable analyses. finally, independence shows a non-significant association in the univariate analysis, but a significant association in the multivariable analysis, suggesting that independence is significantly associated with the response variable after controlling for potential confounding variables.

From Figure 8, the plot can help identify which variable categories are most strongly associated with each dimension. For example, it is apparent from the plot that the categories of the variable namely COM-strongly agree, IAQ-strongly agree, OBJ-strongly agree, and PER-strongly agree are strongly correlated with dimension 2. Similarly, the variables OBJ-strongly disagree, OBJ-agree, IAQ-agree, IND-neutral, PER-agree, BACS-agree, and COM-agree are the most correlated with dimension 1. These insights can be used to interpret the underlying structure of the data.

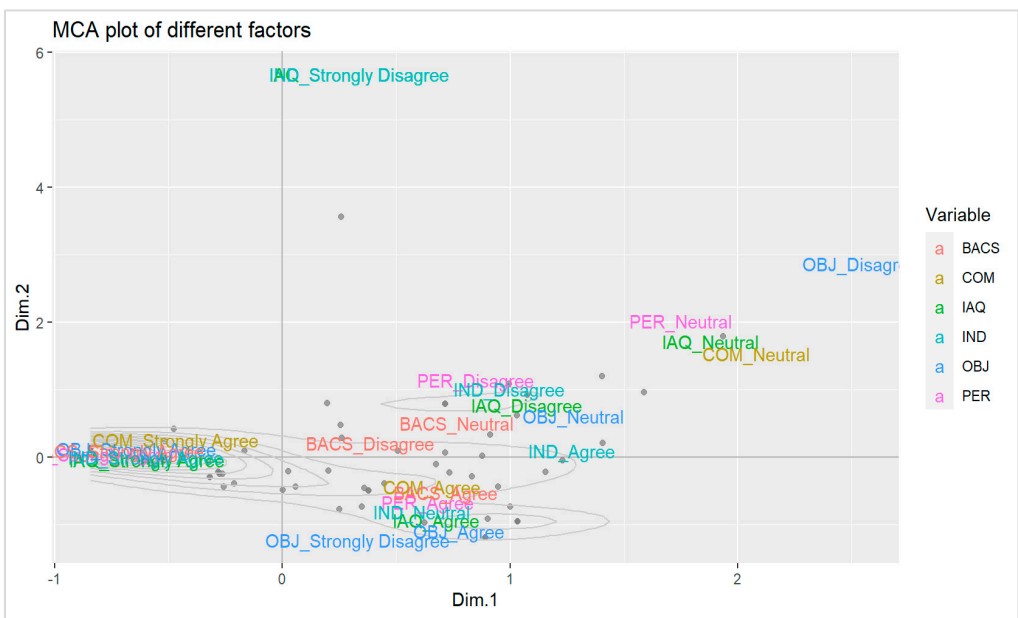

**Figure 8.** Multiple correspondence analysis.

## 5. Conclusions

In a nutshell, internal audit plays a significant and key role to achieve the set goals of the organization. Organizational success depends on the fair and clear system of the internal auditor department along with other elements. Due to its significance internal auditor effectiveness is tricky and critical (Octavia 2013). Evaluation of the proposed hypothesis formulated according to the nature of the current situation in Pakistani commercial banks conclusion is as follows. The main purpose of the research survey is to evaluate the association between the competency of internal audit, objectivity of internal audit, performance of internal auditor, board audit committee support, independence of the internal auditor, and the quality of internal audit. The findings of results revealed that competency, performance, and independence of an internal audit have a significant and positive effect on internal audit quality effectiveness that is closer to the following studies Dellai and Omri (2016); Gramling et al. (2004) Alzeban and Gwilliam (2014); and Baharud-din et al. (2014). The structured survey question revealed the result that internal auditors lack professional certified education (98%) and have less than 10 years of experience in specific internal auditor fields (96%) respectively. So current research is important to attain the attention of the Chief internal auditor and Board audit members of Pakistani Commercial banks to take appropriate measures while hiring auditors because all the progress ultimately depends on the fair audit system.

Concerning this, the purpose of univariate and multivariable logit regression model analysis develops to evaluate the influence of dependent factors on the internal audit quality. Logit regression analysis indicates that competence is the foremost factor evaluate the internal audit quality, followed by performance, competency, and objectivity. The research specifies that Pakistani commercial banks should consider securing the availability of the basic factors to attain the effectiveness of the internal audit.

In summary, the current study's basic aim is to define internal audit characteristics and evaluate the association of intern audit quality effectiveness between factors such as the competence of the internal auditor, independence, objectivity, performance, and board audit committee support. The current research proved the alliance between the abovementioned factors and the effectiveness of internal audit quality in listed commercial banks of Pakistan. From the current study perspective, the stud has some limitations. First and foremost, that mostly focused to assess the listed commercial banks of Pakistan. Secondly, the data available from the perspective of a sample size of the population is small compared to other studies. So, future studies may relate to concentrate factors associated with the internal

audit effectiveness of all financial and non-financial listed companies on the Pakistan stock market and will use a bigger sample population for evaluating and analyzing the outcomes.

**Funding:** This research received no external funding.

**Informed Consent Statement:** Not applicable.

**Data Availability Statement:** The data presented in this study are available on request from the corresponding author.

**Conflicts of Interest:** The author declare no conflict of interest.

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
