# Peer review of "Evaluation of Factors Contributing to the Effectiveness of Internal Audit Quality in Pakistani Commercial Banks"

_ijfs, doi:10.3390/ijfs11040129_

Round 1

Reviewer 1 Report

There is no such a concept in the literature as "effectiveness of internal audit quality" as the Authors of the paper suggest in the title (lines 2-3) and in the introduction (lines 28-30). There are two streams of research in the literature: one relates to internal audit quality (eg. Prawitt, Smith, and Wood 2009; Lin et al. 2011; Abbott et al. 2016; Gramling and Vandervelde 2006; Trotman and Duncan 2017), and the second one that related to internal audit effectiveness (eg. Badara and Saidin, 2013; Endaya and Hanefah, 2013; Lenz and Hahn, 2015; Lenz et al., 2018; Bednarek 2018; Alqudah, Amran, Hassan, 2019; Gamayuni, 2018; Singh, Ravindran, Ganesan, Abbasi, Haron 2021). Authors seem to not be aware of how complex and multidimensional constructs internal audit quality and internal audit effectiveness are and what is the relation between these two concepts. They seem to ignore many previuos studies in the field. They use these terms interchangebly like synonyms (eg. see the title, purpose of the research, table 10, line 117-120) and do not see the need to choose one of them as the dependent variable.

Authors describe the research gap, but not in a good depth. It seems like they try to contribute mainly to the national liturature not international. Authors did not decribe good enough what influential factors of IA effectivess or quality have been already studied in the prevoius studies and which have not. Then another question is how their study relates to the previuos studies (conducted in other countries not only in Pakistan) related to the factors influencing internal audit effectiveness or quality? What is so special about their study in comparison to previous studies that were conducted in the other countries? Can we expect diffrent influential factors of IA effectivess or quality in diffrent sectors or countries (developing/developed) based on the current literature?

Authors do not explain in detail (based on the specific literature) why we can expect that the hypotheses are well formulated. Why can we expect that these particular factors influence internal audit effectiveness or quality in commercial banks in Pakistan?

The methods used are not adequately decribed to replicate the study. There is no information when the data were collected, how the date were collected (via phone, internet, email, in paper send by letter, ect.), what is the number of commercial banks in Pakistan, how many of them took part in the study, how were they sellected to the study. There is some additional information on research methods used in the conlusions section, but it should be moved to the research data section.

There is no clarity what were the questions. For example in regard to question Q1 "The educational level of the internal auditor" it is not clear what was the question and answers as the line 186-187 suggests that the questions were formulated in Likert-type scale (eg. strongly agree - 5, agree - 4, ect).

Independant variable is named as internal audit quality. It consist of six factors. It seems that these factors are related mainly to quality of internal audit products and results. (It is not clear how these questions are formulated in the questionaire.) In the light of the literature (f.e. Gramling and Vandervelde 2006) the Autors of the paper may want to reconsider the name for this variable.

It is not clear why the Authors interpret the results that there is relationship between IAQ and OBJ when it is insignificant with P = 0.895. The same relates to relationship between IAQ and BACS (P = 0.82). See the lines 304-307. Although in the conclusions they admit that this relationship is not signifiacnt (line 323-324), they suggest that objectivity and audit committee support are among factors that influence the internal audit quality (lines 336-338).

In the conclusions one could expect not only practical implications of the study, but also theoretical. There should also be mentioned some limitations of the study.

Author Response

Dear reviewer , we agree with your concers and as per your suggestions we have updated our manuscript please see the attachment. 

Reviewer 2 Report

Article title:  Evaluation of Factors Contributing to the Effectiveness of Internal Audit Quality in Pakistani Commercial Banks

After reviewing this paper carefully, I have some comments below:

- In the introduction, the authors need to clarify research gaps, and contributions of this paper. I haven't seen this in the introduction yet. In addition, the authors need to add a summary of the paper structure at the end of the introduction.

- The authors must restructure the introduction, the content of the paper should not be presented in a list format. The is a sub-section 1.1 but where is 1.2?

- The literature review section of the authors does not meet the requirements. In this section, the authors need to present relevant theories and empirical evidence to propose each of hypothesis. Hypotheses need to be developed separately. I see a lot of previous studies have done this topic but the authors have not reviewed. Therefore, the authors also need to update some recent studies relating to the audit function. I suggest the authors mention and cite some studies such as Dang et al (2022); Nguyen (2022); Muñoz-Izquierdo et al (2019); Dang et al (2021); Nguyen et al (2020) (see references) …

- In the section: research data, the authors need to introduce the process of getting data, filtering data, in which stage is the data taken?. Finally, which companies are included in the data? How many observations are there?

- In the section: research models, the authors need to introduce estimation methods and tests for this estimate.

- What do you mean that you have 2 sections 4 and 5 (Results of the Study and Empirical Results)? I think they should be combined.

- There are 2 sections 5, please check.

- The discussion of the results needs to be done more carefully. Are the results consistent with theories and previous studies? Which hypotheses are supported, and which are not?

- There are some grammatical errors, which the authors need to check carefully.

  References

Dang, V. C. et al (2021). "Internal corporate governance and stock price crash risk: evidence from Vietnam." Journal of Sustainable Finance & Investment: 1-18.

Dang, V. C. et al (2022). "Audit committee characteristics and tax avoidance: Evidence from an emerging economy." Cogent Economics & Finance 10(1): 2023263.

Muñoz-Izquierdo, N., Camacho-Miñano, M. D. M., Segovia-Vargas, M. J., & Pascual-Ezama, D. (2019). Is the external audit report useful for bankruptcy prediction? Evidence using artificial intelligence. International Journal of Financial Studies, 7(2), 20.

Nguyen, Q. et al (2020). "Audit committee structure and bank stability in Vietnam." ACRN Journal of Finance and Risk Perspectives 8(1): 240-255.

Nguyen, Q. K. (2022). "Audit committee structure, institutional quality, and bank stability: evidence from ASEAN countries." Finance Research Letters 46: 102369.

Author Response

(The authors gave the same response as above.)

Reviewer 3 Report

In my opinion this paper include an interesting research topic and also it would be very useful to be published in the context of the recent US banks` bankruptcy. I appreciate the research motivation, the strong literature review and the research objectives, but in my opinion, the use of linear regression analysis is not suitable for this case, because they use Likert scale and variables with no-normal distribution (the parameter for distribution is frequency). In this condition non-parametric tests are recommended and models for variables that have not a normal distribution: please use logit models or multinomial-logit analysis. Also, for descriptives, because you use non-discrete variables (categorical) is not  suitable to have mean and std dev., because if we assume that we have for one item 50% responses for min (1) and 50% responses for max (5), the average would be 3 (total FALSE). In this condition is suitable to use multiple correspondence analysis. For acceptance and publication, authors should reconsider the present analysis with suitable methods and models.  

Author Response

(The authors gave the same response as above.)

Round 2

Reviewer 1 Report

After reviewing this revised manuscript, I have some suggestions as follows:

- There was no available response letter that showed the point-to-point response. Therefore, it took a lot of work to keep track of which points the author edited and which did not.

- I see that some of my suggestions the author still need to implement.

- First of all, there is a need for extensive editing of the English language and style. I have uploaded the manuscript to Grammarly, and it received only 41 out of 100 overall scores, with 654 alerts related to correctness. It is difficult to understand some sentences.

- The author uses the "effectiveness of internal audit quality" concept but does not explain what it means. The author uses such terms as "internal audit quality", "internal audit effectiveness", "internal audit performance", and "effectiveness of internal audit quality" interchangeably (e.g., see lines: 2-3, 7-8, 34-35, 138, 472, 474, 483, 487) and do not see the need to choose one of them as the dependent variable.

- It isn't easy to understand the survey questions. It would be helpful for the reviewer to see the survey questionnaire (in English). For example, does the author ask respondents about their assessment of "the educational level of the internal auditor" or the relationship between "the educational level of the internal auditor" and "internal audit performance"?

Reviewer 2 Report

After reviewing this edit, I have some suggestions as follows:

I can't find the response letter that shows the point-to-point response. Therefore, it is very difficult for me to keep track of which points the author edited and which did not.

I see that some of my suggestions the author has not implemented.

- Literature review and hypothesis development section, the authors have not improved the quality much. The studies I recommend are not up to date yet. Authors need to add citations according to my comment in the previous comment.

- Subsections need to be set up more reasonably. Example: 2.1. Effectiveness of internal audit quality instead of A. Effectiveness...

- The hypothesis development is not convincing. For example, hypothesis H4, H5 is very weakly developed. The authors need to argue based on previous studies. That's why I suggested some of the previous research for the author to update and cite but you didn't do it.

- There are many typographical and grammatical errors that the authors have not yet corrected, for example: (Olga, 2017 described... (page 5).

In summary, the authors need to complete the Literature review and hypothesis development, add citations and correct errors as suggested above.

Reviewer 3 Report

In this form your paper could be accepted for publication.

Author Response

Sir, thank you for your acceptance 

Round 3

Reviewer 2 Report

I read the author's response letter and did not see the authors responding to my comments, the authors just responded to another reviewer's comment. My concern is still not resolved.

Author Response

Dear Editor,

                      I have made changes as per your instructions.

Round 4

Reviewer 2 Report

In this revision, the authors gave satisfactory explanations.